# Does family planning counselling during health service contact improve postpartum modern contraceptive uptake in Ethiopia? A nationwide cross-sectional study

Kalayu Brhane Mruts ![ORCID] ,[1,2] Gizachew Assefa Tessema,[1,3] Jennifer Dunne,[1] Amanuel Tesfay Gebremedhin ![ORCID] ,[1,4] Jane Scott,[1] Gavin F Pereira[1,5]

For numbered affiliations see end of article.

**Correspondence to**
Dr Kalayu Brhane Mruts;
kalayu.brhanemruts@postgrad.curtin.edu.au

## ABSTRACT

**Objective** This study examined the association between family planning counselling receipt during the 12 months preceding the survey and postpartum modern contraceptive uptake in Ethiopia. We hypothesised that receiving family planning counselling either within the community setting by a field health worker or at a health facility by a healthcare attendant during the 12 months preceding the survey improves postpartum modern contraceptive uptake.

**Design** We used a cross-sectional study of the Ethiopian Demographic and Health Survey conducted in 2016.

**Setting** Ethiopia.

**Participants** A total of 1650 women who gave birth during the 12 months and had contact with service delivery points during the 12 months preceding the survey.

**Primary outcome** A weighted modified Poisson regression model was used to estimate an adjusted relative risk (RR) of postpartum modern contraceptives.

**Results** Approximately half (48%) of the women have missed the opportunity to receive family planning counselling at the health service contact points during the 12 months preceding the survey. The postpartum modern contraceptive uptake was 27%. Two hundred forty-two (30%) and 204 (24%) of the counselled and not counselled women used postpartum modern contraceptive methods, respectively. Compared with women who did not receive counselling for family planning, women who received counselling had higher contraceptive uptake (RR 1.32, 95% CI 1.04 to 1.67).

**Conclusion** Significant numbers of women have missed the opportunity of receiving family planning counselling during contact with health service delivery points. Modern contraceptive uptake among postpartum women was low in Ethiopia. Despite this, our findings revealed that family planning counselling was associated with improved postpartum modern contraceptive uptake.

### STRENGTHS AND LIMITATIONS OF THIS STUDY

⇒ This is the first study that examined the association between family planning counselling and postpartum modern contraceptive uptake using large national survey data that enables our finding representative.

⇒ Adjustments were made for potential confounders.

⇒ The family planning counselling and contraceptive uptake were measured using self-reported data with no available previous counselling information that might result in recall bias.

⇒ This is a cross-sectional study that could not assure the causal relationship between family planning counselling and postpartum modern contraceptive uptake.

pregnancies.[1] Countries in sub-Saharan Africa have the lowest postpartum modern contraceptive uptake,[2] accounting for the highest burden of unintended and closely spaced pregnancies.[3–5] Unintended and closely spaced pregnancies are public health issues strongly associated with increased adverse health consequences for the mother and birth outcomes, such as preterm birth, stillbirth, low birth weight, and neonatal and infant mortality.[6–8]

Although modern contraceptive uptake has improved in Ethiopia over the last two decades,[9] the postpartum modern contraceptive uptake is still low,[10 11] resulting in high rates of unintended pregnancy and fertility rates.[9 10 12] For example, a study in Ethiopia using the Ethiopian Demographic and Health Survey (EDHS) 2016 indicated that only 23% of women in the extended postpartum period—the first 12 months following the recent childbirth—used modern contraceptives.[10] Additionally, 27% of Ethiopian women conceive their pregnancy as unintended.[12]

## INTRODUCTION

Delay in the initiation of modern contraceptives in the postpartum period is highly likely to result in unintended and closely spaced

Moreover, it is reported that a short interbirth interval had been associated with stillbirth, preterm birth, neonatal mortality and low birth weight in Ethiopia.[13–15] Provision of family planning counselling during pregnancy, childbirth and postpartum periods is one of the key strategies to improve postpartum modern contraceptive uptake and space subsequent pregnancies.[16] Furthermore, increasing pregnancy spacing could prevent up to 32% of maternal and 21% of child mortalities.[1] In order to improve postpartum contraceptive uptake in low/middle-income countries, the WHO recommends that women receive family planning counselling integrated within maternal and child health services, such as antenatal care (ANC), childbirth, postnatal care (PNC), and well-baby care, immunisation and growth monitoring.[17] To increase access to contraception, the Ethiopian Ministry of Health also recommends that women receive family planning counselling at all the health service contact points, such as health facilities and community settings.[18] The health facilities in the country (public, private or nongovernmental) are expected to provide family planning services integrated with all the health services.[18] Additionally, in the community settings, more than 39 000 health extension workers work across 16 000 villages, mainly in the rural community, to provide health education and counselling on maternal and child health and nutrition, including family planning counselling. Providing appropriate family planning counselling about the potential risks and side effects of modern contraceptive methods is essential for women to avoid misperceptions, increase knowledge and make informed decisions.[19 20]

Despite the strong WHO and Ethiopian Ministry of Health recommendations, Ethiopia lacks research at the national level assessing the association between family planning counselling and postpartum modern contraceptive uptake. Previous studies, conducted with small samples at the district level, have reported mixed evidence.[21–24] This study aims to assess the association between family planning counselling, provided either at the health facility by a health worker or at the community setting by a family planning field worker, and postpartum modern contraceptive uptake in Ethiopia using the most recent national representative EDHS 2016. This study will increase our understanding of the contribution of the family planning counselling provided in Ethiopia on postpartum modern contraceptive uptake. The outcomes of this study will provide decision-makers and programme managers with an evidence base to assess and strengthen the counselling services of family planning. Additionally, it will motivate health workers to provide appropriate family planning information during the counselling sessions.

## METHODS
### Study design and setting
We used cross-sectional data from the nationally representative recent EDHS 2016. Ethiopia is the second

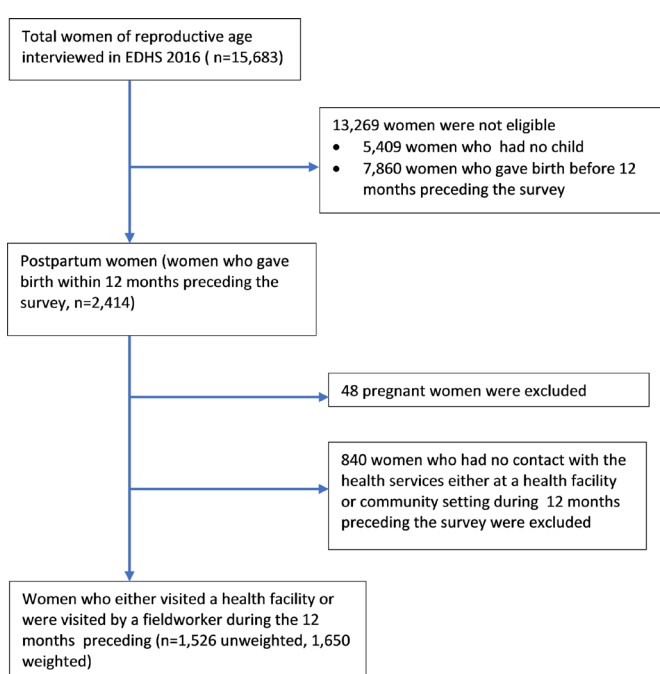

**Figure 1** Participants' selection. The sample weights account for the probability of selection and interview in the EDHS 2016. EDHS, Ethiopian Demographic and Health Survey.

most populous Africa country, with 11 regional states and 2 administrative cities. The EDHS 2016 used a stratified two-stage cluster sampling technique at which each regional state was stratified into urban and rural areas, giving 21 strata. In the first stage, 645 Enumeration Areas (EAs)/clusters, 202 urban areas and 443 rural areas were selected proportional to EA size. In the second stage, 28 households per cluster were randomly selected from the newly created household listing. A detailed description of EDHS 2016 is provided in the full report of the survey.[9]

### Study population
The study population consists of postpartum women who either visited a health facility or were visited by a field worker at the community setting during the 12 months preceding the survey. Women were excluded if they were pregnant again within 12 months after the most recent birth (n=48) and if they had no contact with the health services delivery points during the previous 12 months (ie, neither visited a health facility nor were visited by a field worker at the community setting) (n=840). Thus, the included study population was 1526 women, which corresponded to a weighted sample size of 1650 women (figure 1).

### Variables
#### Outcome variable
The outcome of this study is postpartum modern contraceptive uptake, dichotomised to 'yes' or 'no'. Modern contraceptive uptake was defined when participants were using sterilisation (female or male), condoms (female or male), injectables, oral pills, intrauterine contraceptive

device, implants, lactational amenorrhoea (LAM) and standard calendar method at the time of the survey. Lack of uptake of a modern contraceptive method was defined as lack of use of any contraceptive method or traditional methods, such as abstaining and withdrawals.[9]

### Exposure variable

The main exposure variable for this study was family planning counselling, which is defined as receiving family planning information either by a health service provider at a health facility or a field health worker within a community setting during the 12 months preceding the survey. We categorised the participants into two groups based on their exposure status. The first group consisted of participants who either visited a health facility or were visited by a field health worker in a community setting during the 12 months preceding the survey and received family planning information from a health worker during either of the two contact points and were labelled as 'counselled' for family planning. The second group consisted of participants who either visited a health facility or were visited by a field health worker in a community setting during the 12 months preceding the survey but did not receive family planning information from a health worker at either contact point and were labelled as 'not counselled' for family planning.

### Covariates

After reviewing previous literature,[10] we included sociodemographic characteristics, reproductive histories and health service utilisation covariates for adjustment. The complete list of covariates used for adjustment and their definitions are included in the online supplemental table S1.

### Statistical analysis

We used a generalised linear regression model fitted with Poisson distribution and a log link function to examine the association between receipt of family planning counselling and postpartum modern contraceptive uptake after adjusting for sociodemographic characteristics, reproductive history and health services utilisation. We applied two adjusted models. The first model (model 1) adjusted for all covariates (online supplemental table S1). However, as the adjustment resulted in further loss of precision on the effect estimate, wide CIs, we explored covariates that create a further loss of precision one by one and found to be a place of delivery, PNC visits and postpartum period/interview time. Later, we employed the second model (model 2), adjusting for all covariates, excluding a place of delivery, PNC and postpartum period/interview time. We also run a sensitivity analysis to examine the association of family planning counselling and postpartum modern contraceptive uptake by stratifying the place of delivery, PNC visits and the postpartum period. The samples used in the main analysis and each stratified analysis were included in online supplemental table S2. The $X^2$ test was used to check the distribution of

covariates among counselled and non-counselled women. Multicollinearity among the covariate variables was checked using variance inflation factor. Unadjusted and adjusted relative risks (RRs) with 95% CIs were reported. All analyses accounted for the survey's non-response rate and complex sampling procedure and were undertaken with Stata V.16.[25]

### Patient and public involvement

Patients or the public were not involved in this study.

## RESULTS

This study included 1650 women who had contact with the health service during the 12 months preceding the EDHS (figure 1). The characteristics of the study participants were included in online supplemental table S3. Less than half (48%) of the women reported that they received family planning counselling. Of the women who received counselling, 655 (82%) and 435 (55%) were rural residents and had no education. Of the counselled women, 648 (82%) had attended at least one ANC visit during their last pregnancy. However, 401 (52%) and 561 (72%) were women who gave birth at home and did not attend PNC visits, respectively. Except for maternal age, parity, distance to a health facility, health insurance coverage and receipt of media-delivered family planning messages, all women's characteristics had similar distributions among those counselled and non-counselled. Of the counselled women, 361 (45%) were 30 years and above, whereas of the non-counselled women, 269 (21%) were 30 years and above. In reverse to this, of the counselled and non-counselled women, 231 (29%) and 318 (37%) were less than 25 years, respectively. Of those who received counselling from a health worker, 289 (36%) received family planning messages from the media, while of those who did not receive counselling from a health worker, only 208 (24%) received family planning messages from the media. Of the study women <6 months post partum, 661 (84%) were postpartum amenorrhoeic, but only 473 (60%) met the LAM criteria. Indeed, more than three-quarters of women <6 months post partum in our study were not using modern contraceptive methods (table 1).

There were 445 (27%) participants who used modern contraceptive methods at the survey time. The most used modern contraceptive method was injectable (18.4%%), followed by implants (4.7%). Two hundred forty-two (30%) and 204 (24%) of the counselled and non-counselled women used postpartum modern contraceptive methods, respectively (table 2).

Before adjustment, family planning counselling was associated with 27% higher postpartum modern contraceptive uptake (RR 1.27, 95% CI 1.00 to 1.62). Moreover, the adjusted model (model 1) indicated that the risk of postpartum modern contraceptive uptake was 25% higher among women who have been counselled for family planning than those who did not receive

**Table 1** Characteristics of the study participants by the status of family planning counselling in Ethiopia 2016 (n=1650)

| Variable | Category | Family planning counselling | |
| --- | --- | --- | --- |
| | | Not counselled (%) N=856 (51.9) | Counselled (%) N=794 (48.1) |
| Household wealth index | Lowest | 162 (18.9) | 155 (19.6) |
| | Middle | 379 (44.3) | 306 (38.5) |
| | Highest | 315 (36.8) | 333 (41.9) |
| Residence | Urban | 116 (13.5) | 139 (17.5) |
| | Rural | 740 (86.5) | 655 (82.5) |
| Region | Tigray | 67 (7.8) | 93 (11.8) |
| | Amhara | 189 (22.1) | 174 (22.0) |
| | Oromia | 344 (34.4) | 299 (37.6) |
| | SNNPR | 165 (19.3) | 175 (22.0) |
| | Addis Ababa | 30 (3.5) | 25 (3.2) |
| | Others | 61 (7.2) | 27 (3.5) |
| Women's age | <25 years | 318 (37.2) | 231 (29.1) |
| | 25–29 years | 269 (31.4) | 202 (25.4) |
| | ≥30 years | 269 (21.4) | 361 (45.5) |
| Women's educational level | No education | 447 (52.2) | 435 (54.8) |
| | Primary | 314 (36.7) | 271 (34.2) |
| | Secondary and above | 95 (11.1) | 87 (11.0) |
| Women's current job | Not working | 542 (63.3) | 454 (57.2) |
| | Working | 314 (36.7) | 340 (42.8) |
| Religion | Christian | 477 (55.7) | 505 (63.6) |
| | Muslim | 362 (42.3) | 273 (34.4) |
| | Others | 17 (2.0) | 16 (2.0) |
| Women's decision-making autonomy | No autonomy | 308 (39.0) | 262 (34.1) |
| | Having autonomy | 481 (61.0) | 506 (65.9) |
| Parity | Primiparity | 240 (28.5) | 165 (20.8) |
| | Multiparity | 431 (50.4) | 419 (52.8) |
| | Grand multiparity | 181 (21.1) | 210 (26.4) |
| Wantedness of the last pregnancy | Unintended | 232 (27.1) | 217 (27.3) |
| | Intended | 624 (72.9) | 577 (72.7) |
| Sex of the last child | Male | 419 (49.0) | 402 (50.7) |
| | Female | 437 (51.0) | 391 (49.3) |
| Survival status of the last child | Died | 20 (2.3) | 21 (2.7) |
| | Alive | 836 (97.7) | 772 (97.3) |
| ANC visits for the last pregnancy | No | 200 (23.4) | 146 (18.4) |
| | Yes | 654 (76.6) | 648 (81.6) |
| Place of delivery for the last birth | Home | 472 (56.2) | 401 (51.6) |
| | Health facility | 368 (43.8) | 376 (48.4) |
| PNC visits following the last birth | No | 642 (75.9) | 561 (71.9) |
| | Yes | 180 (24.1) | 199 (28.1) |
| Interview time/postpartum period | 0–6 months | 468 (54.7) | 466 (58.7) |
| | 7–12 months | 388 (45.3) | 328 (41.3) |
| Fertility intention | No more | 288 (33.8) | 340 (43.0) |
| | Have another | 565 (66.2) | 451 (57.0) |

**Table 1** Continued

| Variable | Category | Family planning counselling | |
| | | Not counselled (%) N=856 (51.9) | Counselled (%) N=794 (48.1) |
| --- | --- | --- | --- |
| Receiving family planning messages from media | No | 648 (75.6) | 505 (63.6) |
| | Yes | 208 (24.4) | 289 (36.4) |
| Have health insurance | No | 834 (97.4) | 737 (92.9) |
| | Yes | 22 (2.6) | 56 (7.1) |
| Distance to a health facility | A big problem | 501 (58.6) | 409 (51.6) |
| | Not a big problem | 355 (41.4) | 385 (48.4) |

ANC, antenatal care; PNC, postnatal care; SNNPR, South, Nation, Nationalities and Peoples' Region.

counselling, although point estimates were imprecise (RR 1.32; 95% CI 1.04 to 1.67). It was observed that place of delivery, PNC visits and the postpartum period were the covariates that resulted in further loss of precision. The result of the adjusted model (model 2), excluding place of delivery, PNC visits and postpartum period/interview time, also showed an association between family planning counselling and postpartum modern contraceptive uptake (RR 1.25; 95% CI 0.99 to 1.60) (table 3).

We performed a sensitivity analysis to further look at the effect of family planning counselling on postpartum contraceptive uptake by stratifying the place of delivery by home versus facility birth status, PNC visits as yes versus no, and the postpartum period into 0–6 vs 7–12 months. Although the effect estimates were imprecise (wide CIs), point estimates indicated that family planning counselling improved contraceptive uptake regardless of the place of delivery and PNC visits. However, the group for which counselling was observed to be associated with improved contraceptive uptake gave birth 7–12 months prior to undertaking the survey.

## DISCUSSION

This study evaluated the association between counselling for family planning and postpartum modern contraceptive uptake in Ethiopia using the EDHS 2016.

Our finding revealed that family planning counselling during the 12 months preceding the survey was associated with postpartum modern contraceptive uptake. This was supported by previous studies.[16 21 22] This was inconsistent with some previous studies from Ethiopia, which indicated

a lack of association between family planning counselling and postpartum modern contraceptive uptake.[23 26 27] One reason for the inconsistency might be due to the differences in the quality of the family planning counselling. Although the previous studies conducted in Ethiopia on family planning counselling and postpartum contraceptive uptake did not describe the quality of the counselling, the quality can create a difference in adopting the contraceptive methods.[28 29] For example, a study conducted in Uttar Pradesh, India showed that receiving higher quality family planning counselling improves the use of modern contraceptive methods.[28] Sociocultural differences might also explain the effect of family planning counselling on postpartum modern contraceptive uptake inconsistencies, as Ethiopia is multicultural with varied religious beliefs

**Table 2** The proportion of postpartum modern contraceptive use among counselled and not counselled women in Ethiopia 2016

| Family planning counselling status | Postpartum modern contraceptive uptake | |
| | Yes (%) | No (%) |
| --- | --- | --- |
| Counselled | 242 (30) | 553 (70) |
| Not counselled | 204 (24) | 652 (76) |

**Table 3** Association between family planning counselling and postpartum modern contraceptive uptake stratified by place of delivery, PNC visits and interview time post partum in Ethiopia 2016

| | Unadjusted | Adjusted* |
| --- | --- | --- |
| All women | 1.27 (1.00 to 1.62) | 1.25 (0.99 to 1.60) |
| | Place of delivery | |
| Home delivery | 1.35 (0.86 to 2.13) | 1.33 (0.83 to 2.12) |
| Facility delivery | 1.18 (0.90 to 1.54) | 1.27 (0.96 to 1.67) |
| | PNC visits | |
| Had PNC visits | 1.05 (0.73 to 1.51) | 1.25 (0.85 to 1.83) |
| Had no PNC visits | 1.39 (1.02 to 1.90) | 1.27 (0.92 to 1.74) |
| | Interview time post partum | |
| 0–6 months post partum | 1.00 (0.66 to 1.52) | 0.90 (0.27 to 2.96) |
| 7–12 months post partum | 1.54 (1.17 to 2.03) | 1.60 (1.23 to 2.07) |

*Adjusted for household wealth status, residence, maternal age, maternal education, maternal employment, women's decision-making autonomy, religion, region, parity, survival status of the last child, sex of last child, wantedness of last pregnancy, fertility intention, ANC, distance to health facility, health insurance coverage and receiving of family planning messages from media. ANC, antenatal care; PNC, postnatal care.

that may contribute to differences in perceptions and beliefs regarding contraceptive methods.[29 30] Therefore, even if women obtained counselling on family planning, this may not necessarily be translated to contraceptive use. Notable examples include sociocultural influences, such as gender inequality of power in decision-making or religious prohibitions to use contraceptive methods.[29 31]

Our study also indicated that only half (48%) of the women received family planning counselling. This is slightly higher than the previous studies in Ethiopia, which indicated that only 28%–35% of women were provided family planning counselling.[32 33] This discrepancy might be due to the difference in family planning counselling definition. Unlike the previous studies that used only facility-based counselling, our definition includes counselling provided in facility and/or community settings. Nonetheless, the result indicates that despite the fact that family planning services are integrated with maternal and child health services at the policy level,[18 30] there remains a gap in the universal adoption of the recommendation by healthcare providers to counsel family planning, resulting in a higher proportion of women who missed the opportunity to be counselled. Although further research is needed on the barriers to the missed opportunity of providing family planning counselling while visiting service delivery points, previous qualitative research indicated that lack of awareness by the service providers of the national guidelines and the lack of the guidelines in the health facilities were the main barriers to not adhering to it.[34] The structural barriers, such as shortage of infrastructure and equipment and the unbalanced client-to-provider ratios within the Ethiopian healthcare system,[35] might contribute to the high missed opportunity to provide family planning counselling.

Adjusted point estimates differed after additional adjustment for the place of delivery, PNC and the postpartum period/interview time, implying that the pathway through which family planning counselling affects contraceptive uptake may act through these variables. We conducted additional sensitivity analyses to investigate further these pathways, which indicated that family planning counselling improved postpartum contraceptive uptake regardless of the place of delivery and PNC visits. However, we also observed that counselling was associated with improved postpartum contraceptive uptake only for women interviewed at least seven months after giving birth. Delivery at a health facility or receipt of PNC that includes family planning counselling may not translate an increased likelihood of modern contraceptive uptake if women believe that they would be free of the risk of pregnancy while amenorrhoeic following delivery. Ethiopian women may consider resumption menses as critical timing for starting contraception in the postpartum period.[27] Ethiopian women may also believe that they are not at risk of pregnancy in the first 6 months after giving birth if they are breast feeding their baby, whether or not breast feeding is exclusive.[36] The findings of stronger associations between family planning counselling and

postpartum modern contraceptive uptake among women 7–12 months post partum could also be because there might be a greater proportion of women who have re-engaged in sexual activity during this period than in the first 6 months.

Overall, our findings highlighted the importance of integration of family planning counselling with other health services, particularly with maternal and child health services. This helps dispel misperceptions and enable women to make informed decisions that would significantly reduce missed opportunities, thereby improving family planning counselling coverage and the uptake of postpartum modern contraception. Hence, the Ethiopian government and partner organisations with the remit of family planning, maternal and child health should work to translate the plans documented in the National Family Planning Guideline and the National Reproductive Health Strategy.

This is the first study that assessed the association between family planning counselling and postpartum modern contraceptive uptake at the national level and provides insight for government and partner organisations on the gaps in the provision of family planning counselling. However, this study has several limitations that need to be considered when interpreting our findings. The type and frequency of family planning information and the health worker who provided the counselling were not collected within the EDHS. Consequently, we were unable to examine the quality of counselling provided. Meanwhile, as the women were asked to provide family planning information for the 12 months prior to undertaking the survey, information before this time might have contributed to the association between family planning counselling and postpartum modern contraceptive uptake. Furthermore, although the data were based on self-reported information that occurred in the preceding 12 months, we cannot completely rule out the influence of recall bias. Lastly, due to the nature of the cross-sectional study design, it is not easy to determine the causal relationship between family planning counselling and postpartum modern contraceptive uptake.

## CONCLUSION

Significant numbers of postpartum women did not receive family planning counselling despite contact with health service contact points during the preceding 12 months. We found that postpartum modern contraceptive uptake was low. Nonetheless, our findings reveal that family planning counselling is associated with improving postpartum modern contraceptive uptake in Ethiopia. Hence, besides preparing guidelines and working documents, the government should work to fully implement the integration of family planning counselling at all levels of service delivery points. Additionally, all service providers should deliver family planning counselling by complying with the national guideline, and postpartum counselling should stress the early initiation of modern

contraceptives after giving birth. On top of these, continuous monitoring and evaluation of family planning counselling integration would help to reduce the missed opportunities and improve contraceptive uptake.

**Author affiliations**
[1]School of Population Health, Curtin University, Perth, Western Australia, Australia
[2]School of Public Health, Debre Berhan University, Debre Berhan, Amhara, Ethiopia
[3]School of Public Health, The University of Adelaide, Adelaide, South Australia, Australia
[4]Wesfarmers Centre for Vaccine and Infectious Disease, Telethon Kids Institute, University of Western Australia, Perth, Western Australia, Australia
[5]School of Population, Centre for Fertility and Health (CeFH), Norwegian Institute of Public Health, Oslo, Norway

**Acknowledgements** We are grateful to the DHS programme for providing access to the Ethiopian Demographic and Health Survey data.

**Contributors** KBM—conceptualisation, formal analysis, data curation, writing (original draft), writing (review and editing), visualisation. GAT, JD, ATG, JS, GFP—writing (review and editing), visualisation. KBM is the author acting as guarantor.

**Funding** The authors did not receive financial or material support for this study from any public or private organisation. However, GFP and GAT are investigators of Australian National Health and Medical Research Council (project grants #1099655 and #1173991 for GFP, and #1195716 for GAT). GFP also received institutional funding from the WA Health and Artificial Intelligence Consortium and is supported by the Research Council of Norway through its Centres of Excellence funding scheme (#262700 for GFP).

**Competing interests** None declared.

**Patient and public involvement** Patients and/or the public were not involved in the design, or conduct, or reporting, or dissemination plans of this research.

**Patient consent for publication** Not required.

**Ethics approval** The EDHS data were collected after ethical consent was sought from the Science and Technology Ministry, Ethiopia. Moreover, we obtained permission to use the EDHS dataset from the DHS programme, which required no further ethical approval.

**Provenance and peer review** Not commissioned; externally peer reviewed.

**Data availability statement** Data are available in a public, open access repository. Data are available in public open access in the DHS programme repository (https://dhsprogram.com/). Registration is required to access the dataset.

**ORCID iDs**
Kalayu Brhane Mruts http://orcid.org/0000-0001-5655-5636
Amanuel Tesfay Gebremedhin http://orcid.org/0000-0003-2459-1805

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
