## [Reviewer comments · BMJ Open]

ARTICLE DETAILS

TITLE (PROVISIONAL)	Does family planning counselling during health service contact improve postpartum modern contraceptive uptake in Ethiopia? A nationwide cross-sectional study
AUTHORS	Mruts, Kalayu Brhane; Tessema, Gizachew; Dunne, Jennifer; Gebremedhin, Amanuel Tesfay; Scott, Jane; Pereira, G.F

VERSION 1 – REVIEW

REVIEWER	Dev, Rubee Sun Yat-Sen University
REVIEW RETURNED	08-Jan-2022

GENERAL COMMENTS	This manuscript aims to look at the association between family planning counseling and uptake of modern contraceptives by postpartum women in Ethiopia. While this laudable approach could offer an updated evidence on the contraceptives uptake, below are some of the suggested ways in which the paper could be improved and hope that the authors find these suggestions useful. 1. Exposure variable: It needs to be clear, whom did they receive family planning counseling from. Did they receive counseling from the same cadre of health care professional? Counseling received through the media will be different than those received from the health care professional. Also, what were the components of counseling?2. Outcome variable, Page 7, Line 36-47: Definition of modern contraceptive uptake requires a citation. Also, the definition used in the manuscript seems to be mixed with the traditional methods such as standard calendar method that might not fit under the definition of modern contraceptives.3. Covariates could be listed after mentioning the explanatory and outcome variables in the manuscript.4. Page 9, Line 10: The sentence seems to end abruptly here and might require paraphrasing to make it clearer.5. Discussion: Should start with the key findings of the study rather than comparing and contrasting with other studies directly.6. This manuscript requires proof-reading for incomplete sentences in multiple instances. E.g, Page 14, Line 28-30; Page 15, Line 22- it should be family planning counseling rather than family counseling; Page 15, Line 50- at least seven months after...etc.7. Authors discuss about the quality of the counseling as a limitation for other studies. Quality aspect has not been examined in the given study as well. How would authors justify this?
---

REVIEWER	Segni, Mesfine Arsi University, Public Health
-----------------	--

REVIEW RETURNED	28-Feb-2022
-------------

GENERAL COMMENTS	The work is interesting where the Unmet need for family planning is significant in developing countries like Ethiopia. It adds input to the country policy on the modern family planning service delivery in facilities especially to check the quality and the allocated time for counseling by providers at different levels in the country. But I have two concerns about the paper is reporting the effect measurement, as known the demographic surveys are a one-point time study(cross-sectional study) as the outcome is not measured after following, the appropriate measure of association is ODDS RATIO(OR). As far as my knowledge the reported Relative Risk is appropriate it is better to change to OR. The second concern is about the region category, currently, some regions Like SNNP was split into three regions, If possible try to split it and see the various if the data allows. Actually, the data were collected before departing the regions. The rest is OK.
--

REVIEWER	Beyene, Markos Wollega University
REVIEW RETURNED	15-Mar-2022

GENERAL COMMENTS	Title and Abstract Title is well stated and informative Background is not discussed separately and background related issues are discussed under objective. Therefore I suggest the authors to have background and objectives separately. Methods section of abstract could not be segregated as: design, participants or outcome. Conclusion section of abstract is well stated Strength and limitation section is well informative Introduction Introduction section coherent all issue need to be addressed in this section has been mentioned. But there are issues need revision. The sentence on line 9 and 10 should have reference The third paragraph could have been about risk factors associated with the problem under study. This paragraph will be placed between the second paragraph and the paragraph dealing with intervention. Methods Line 9: the study design and setting start as: "we used....." Better to avoid the word 'we' and rephrase it again. Line 10-23 paragraphs are with no reference. Variable section is well stated and informative Analysis Line 52 rephrases the sentence and avoids "we". Page 8, line 33-35 is not clear Results Avoid using 'of the" at the beginning of the sentences redundantly throughout the result section. Tables should be self explanatory, should answer what, when, where ..?) Discussion Line 28-30 need language editions The discussion and the justification given for possible reason for the difference seen between this study and other study is almost about quality which was not the objective of this study. The last two paragraphs of the discussion looks like conclusion
---

	Conclusion is well stated Tables and figures All figures, tables and supplementary tables should self explanatory
--	--

VERSION 1 – AUTHOR RESPONSE

Reviewer: 1

Dr. Rubee Dev, Sun Yat-Sen University

Comments to the Author:

This manuscript aims to look at the association between family planning counseling and uptake of modern contraceptives by postpartum women in Ethiopia. While this laudable approach could offer an updated evidence on the contraceptives uptake, below are some of the suggested ways in which the paper could be improved and hope that the authors find these suggestions useful.

Q1: Exposure variable: It needs to be clear, whom did they receive family planning counseling from. Did they receive counseling from the same cadre of health care professional? Counseling received through the media will be different than those received from the health care professional. Also, what were the components of counseling?

Reply: Thank you for your question. As we describe in the definition of family planning counselling on page 7, lines 7-30, women have received family planning counselling either within the community (e.g., house-to-house visit) by a field health worker or at a health facility by a healthcare attendant. For better clarity, we have now added counselling “*from a health worker*” in the definition of family planning counselling (see on Exposure variable sub-section page 8, line 3-14).

We did not include the family planning information received from the media in our definition. However, family planning information from media was adjusted in our analyses. The EDHS interview did not collect the components of the counselling but this has been acknowledged in the limitations section (page 16, lines 23-25)

Q2. Outcome variable, Page 7, Line 36-47: Definition of modern contraceptive uptake requires a citation. Also, the definition used in the manuscript seems to be mixed with the traditional methods such as standard calendar method that might not fit under the definition of modern contraceptives.

Reply: Thank you for your good question. As the statement on page 7, lines 21-24 and page 8, lines 1-2, is about the definition of modern contraceptives, we have already cited a reference (see reference 8 in the revision) . As our data source is from the EDHS, we used the definition used by the EDHS ¹. The World Health Organization also similarly defined modern contraceptive methods (<https://www.who.int/data/gho/indicator-metadata-registry/imr-details/3334>).

Of course, some scholars do not adopt this definition and exclude the standard calendar method from the definition of modern contraceptive methods². However, using the same definition with the EDHS

reports would help our comparability of our findings with the EDHS report, particularly those relating to postpartum modern contraceptive uptake, and other previous studies that used similar definitions^{3 4}

Q3. Covariates could be listed after mentioning the explanatory and outcome variables in the manuscript.

Reply: Thank you for your suggestion. We have made the suggested changes (page 7, lines 20-24 and page 8, lines 1-2 & 16-20)

Q4. Page 9, Line 10: The sentence seems to end abruptly here and might require paraphrasing to make it clearer.

Reply: Thank you for your suggestion. For better clarity, we have added “*from a health worker*”

*...Of those who received counselling **from a health worker**, 289 (36%) received family planning messages from the media, while of those who did not receive counselling **from a health worker**, only 208 (24%) received family planning messages from the media.*

Q5. Discussion: Should start with the key findings of the study rather than comparing and contrasting with other studies directly.

Reply: Thank you for your suggestions. We have now started the Discussion with the key findings as per the reviewer’s suggestion. Specifically, we have edited the description on page 14, line 9 and page 15 lines 1-2

Q6. This manuscript requires proof-reading for incomplete sentences in multiple instances. E.g, Page 14, Line 28-30; Page 15, Line 22- it should be family planning counseling rather than family counseling; Page 15, Line 50- at least seven months after...etc.

Reply: Thank you for your comments. We believe the statement on page 14, lines 28-30 is complete. “it” at the end refers to the national family planning guideline. We used “it” instead of repeatedly describing the national family planning guideline.

We did not find an incomplete statement on page 15, line 22; instead, the incompleteness is on page 14, line 22, and we have edited as suggested.

*...is needed on the barriers to the missed opportunity of providing family **planning** counselling....*

For the statement on page 15, line 50, we have modified it as follows

..The findings of stronger associations between family planning counselling and postpartum modern contraceptive uptake among women 7-12 months postpartum could also be because there might be a greater proportion of women who have re-engaged in sexual activity during this period than in the first six months.

Q7. Authors discuss about the quality of the counseling as a limitation for other studies. Quality aspect has not been examined in the given study as well. How would authors justify this?

Reply: Thank you for your good question. Quality of family planning counselling could encompass many aspects. Due to unavailability in the dataset, we were not able to elucidate the role of type and frequency of family planning information on improving postpartum modern contraceptive uptake and this was described in the limitations on (page 16, lines 23-25). Moreover, as many health workers can provide counselling in Ethiopia, we were not able to determine which health worker was influential in adopting the postpartum modern contraceptive methods.

Reviewer: 2

Dr. Mesfine Segni, Arsi University

Comments to the Author:

The work is interesting where the Unmet need for family planning is significant in developing countries like Ethiopia. It adds input to the country policy on the modern family planning service delivery in facilities especially to check the quality and the allocated time for counseling by providers at different levels in the country.

Q1: But I have two concerns about the paper is reporting the effect measurement, as known the demographic surveys are a one-point time study (cross-sectional study) as the outcome is not measured after following, the appropriate measure of association is ODDS RATIO (OR). As far as my knowledge the reported Relative Risk is appropriate it is better to change to OR.

Reply: Thank you for your good question. In the case of rare disease, or if the prevalence of the outcome of interest is less than 10%, the OR and RR are approximately equal, and the OR can be used to estimate the RR in a cohort or cross-sectional studies. However, as the prevalence of the outcome of interest is 27% in our case, OR overestimates the RR⁵⁻⁷. Additionally, OR is prone to misinterpretation. Hence, it is recommended to use RR in a cohort or cross-sectional studies^{5 6}.

Q2: The second concern is about the region category, currently, some regions like SNNP was split into three regions, if possible try to split it and see the various if the data allows. Actually, the data were collected before departing the regions. The rest is OK.

Reply: As it is mentioned, the EDHS 2016 was collected before the establishment of new regional States, which were parts of the SNNPR, in Ethiopia. So, we could not create and analyse the data by the new governmental structure. However, we have updated the statement "nine regional states" into "11 regional states" considering the newly established regional states in the Study Design and Setting sub-section starting at page 7, line 4.

Reviewer: 3

Dr. Markos Beyene, Wollega University

Comments to the Author:

the author needs to revise based on the given comments from the title to the end. the manuscript will have great input for the scientific community if published in this journal

Title and Abstract

Title is well stated and informative

Q1. Background is not discussed separately and background related issues are discussed under objective. Therefore I suggest the authors to have background and objectives separately.

Reply: Thank you for your suggestion. The BMJ Open author guideline recommends “ a clear statement of the main study aim and major hypothesis/research question”. Now, we included only the objective and research hypothesis to adhere to the guideline (see the objective section of the abstract page 2, lines 2-7).

Q2. Methods section of abstract could not be segregated as design, participants or outcome.

Conclusion section of abstract is well stated

Reply: Thank you for your suggestion. As we described above, we separated the methods section into design, participants and outcome based on the BMJ Open author guidelines.

Strength and limitation section is well informative

Introduction

Q3. Introduction section coherent all issue need to be addressed in this section has been mentioned. But there are issues need revision.

The sentence on line 9 and 10 should have reference

Reply: Thank you for your suggestion. The sentence has been removed in the revised version of the manuscript.

Q4. The third paragraph could have been about risk factors associated with the problem under study. This paragraph will be placed between the second paragraph and the paragraph dealing with intervention.

Reply: Thank you for your suggestions. We have edited the introduction section. In the first and second paragraphs, we describe how common the problem (postpartum modern contraceptive uptake) is in low-income settings and Ethiopia. The third paragraph states the role of family planning counselling during the maternal continuum of care on improving the postpartum modern contraceptive uptake in low-income settings, including Ethiopia. The fourth paragraph describes the gap and significance of the study. It will not be meaningful to describe the influence of family planning counselling on postpartum modern contraceptive uptake before first describing the extent of postpartum modern contraceptive uptake in SSA and Ethiopia and its impact.

Methods

Q5. Line 9: the study design and setting start as: “we used.....” Better to avoid the word ‘we’ and rephrase it again.

Reply: Thank you for your suggestion. It is written in active voice as using active or passive voice has no problem.

Q6. Line 10-23 paragraphs are with no reference.

Reply: Thank you for your comment. The description on lines 10-23 is taken from the EDHS report, and we have now cited the reference at the end of the paragraph.

Variable section is well stated and informative

Q7. Analysis Line 52 rephrases the sentence and avoids “we”.

Reply: Thank you for your suggestion. It is written in active voice as using active or passive voice has no problem.

Q8. Page 8, line 33-35 is not clear

Reply: Thank you for your comment. Including patient and public involvement is one component that needs to be included in the manuscript as suggested in the author guidelines of the journal. As we conducted our analysis from secondary data of EDHS 2016, there were no patients or public involved in the design or implementation of this study.

Results

Q9. Avoid using ‘of the’ at the beginning of the sentences redundantly throughout the result section.

Reply: Thank you for your comment. On page 9, lines 18-25 and page 10, lines 1-7 are about the women’s characteristics by family planning counselling status, we used “of the” to differentiate the counselled vs not counselled women. Reporting cross-tabulation is not straightforward as reporting the simple frequency. Unless we use “of the” the information would be misleading readers.

Q10. Tables should be self explanatory, should answer what, when, where ..?)

Reply: Thank you for the suggestion. Now we have updated the tables in response to your comment.

Table 1. Characteristics of the study participants by the status of family planning counselling in Ethiopia 2016 (n=1,650)

Discussion

Q11. Line 28-30 need language editions

Reply: Thank you for your suggestion. Now we have edited these lines.

*One reason for the inconsistency might be due to the differences **in the** quality of the family planning counselling.*

Q12. The discussion and the justification given for possible reasons for the difference seen between this study and other studies is almost about quality which was not the objective of this study.

Reply: Thank you for the question. We did not mention the quality of family planning as the only reason, but we also included sociocultural factors as additional justification for the discrepancies between our study finding and previous studies. They are reasons that could possibly justify the inconsistencies between our research finding(s) and other studies.

Q13. The last two paragraphs of the discussion looks like conclusion

Reply: Thank you for your comment. The last two paragraphs of the discussion describe the strengths and limitations and the implications of this study in implementing the national guidelines and strategies for improving family planning counselling. None of our study findings are described in the two last paragraphs.

Conclusion is well stated

Tables and figures

Q14. All figures, tables and supplementary tables should self explanatory

Reply: Thank you for your suggestions. We have edited all the table labels accordingly

Table 2. Characteristics of the study participants by the status of family planning counselling in Ethiopia 2016 (n=1,650)

Table 2. The proportion of postpartum modern contraceptive use among counselled and not counselled women in Ethiopia 2016

Table 3. Association between family planning counselling and postpartum modern contraceptive uptake stratified by place of delivery, PNC visits and interview time postpartum in Ethiopia 2016.

Table S2. Sample size for the adjusted and unadjusted models among all and women stratified by place of delivery, PNC visits and interview time postpartum in Ethiopia 2016

Table S3. Sociodemographic characteristics, reproductive histories, and health service utilisation of the study participants in Ethiopia 2016 (n=1,650)

References

1. Central Statistical Agency - CSA/Ethiopia, ICF. Ethiopia Demographic and Health Survey 2016. Addis Ababa, Ethiopia: CSA and ICF, 2017.
2. Hubacher D, Trussell J. A definition of modern contraceptive methods. *Contraception* 2015;92(5):420-21. doi: 10.1016/j.contraception.2015.08.008
3. Dagnew GW, Asresie MB, Fekadu GA, et al. Modern contraceptive use and factors associated with use among postpartum women in Ethiopia; further analysis of the 2016 Ethiopia demographic and health survey data. *BMC Public Health* 2020;20(1):661. doi: 10.1186/s12889-020-08802-6
4. Tessema GA, Mekonnen TT, Mengesha ZB, et al. Association between skilled maternal healthcare and postpartum contraceptive use in Ethiopia. *BMC Pregnancy and Childbirth* 2018;18(1):172. doi: 10.1186/s12884-018-1790-5
5. Schmidt CO, Kohlmann T. When to use the odds ratio or the relative risk? *International Journal of Public Health* 2008;53(3):165-67. doi: 10.1007/s00038-008-7068-3
6. Wilber ST, Fu R. Risk Ratios and Odds Ratios for Common Events in Cross-sectional and Cohort Studies. *Academic Emergency Medicine* 2010;17(6):649-51. doi: 10.1111/j.1553-2712.2010.00773.x
7. Viera AJ. Odds ratios and risk ratios: what's the difference and why does it matter? *South Med J* 2008;101(7):730-4. doi: 10.1097/SMJ.0b013e31817a7ee4